# Characterizing the Role of Biologically Relevant Fluid Dynamics on Silver Nanoparticle Dependent Oxidative Stress in Adherent and Suspension In Vitro Models

**DOI:** 10.3390/antiox10060832

**Published:** 2021-05-23

**Authors:** Katherine E. Burns, Robert F. Uhrig, Maggie E. Jewett, Madison F. Bourbon, Kristen A. Krupa

**Affiliations:** Department of Chemical and Materials Engineering, University of Dayton, Dayton, OH 45469-0256, USA; burnsk7@udayton.edu (K.E.B.); uhrigr1@udayton.edu (R.F.U.); jewettm3@udayton.edu (M.E.J.); mbourbon1@udayton.edu (M.F.B.)

**Keywords:** silver nanoparticle, reactive oxygen species, cytotoxicity, dynamic flow, p53, NFκB, cytokine secretion

## Abstract

Silver nanoparticles (AgNPs) are being employed in numerous consumer goods and applications; however, they are renowned for inducing negative cellular consequences including toxicity, oxidative stress, and an inflammatory response. Nanotoxicological outcomes are dependent on numerous factors, including physicochemical, biological, and environmental influences. Currently, NP safety evaluations are carried out in both cell-based in vitro and animal in vivo models, with poor correlation between these mechanisms. These discrepancies highlight the need for enhanced exposure environments, which retain the advantages of in vitro models but incorporate critical in vivo influences, such as fluid dynamics. This study characterized the effects of dynamic flow on AgNP behavior, cellular interactions, and oxidative stress within both adherent alveolar (A549) and suspension monocyte (U937) models. This study determined that the presence of physiologically relevant flow resulted in substantial modifications to AgNP cellular interactions and subsequent oxidative stress, as assessed via reactive oxygen species (ROS), glutathione levels, p53, NFκB, and secretion of pro-inflammatory cytokines. Within the adherent model, dynamic flow reduced AgNP deposition and oxidative stress markers by roughly 20%. However, due to increased frequency of contact, the suspension U937 cells were associated with higher NP interactions and intracellular stress under fluid flow exposure conditions. For example, the increased AgNP association resulted in a 50% increase in intracellular ROS and p53 levels. This work highlights the potential of modified in vitro systems to improve analysis of AgNP dosimetry and safety evaluations, including oxidative stress assessments.

## 1. Introduction

Due to their physicochemical properties, nanoparticles (NPs) possess unique behavioral traits, such as augmented reactivity, increased transport potential, and distinctive optical signatures. The utilization of these properties and behavioral characteristics has resulted in the incorporation of NPs into hundreds of consumer products and applications [1]. To date, silver NPs (AgNPs) are the most predominantly employed, with applications spanning the medical, electrical, industrial, consumer goods, and health care sectors [2,3]. Owing to their robust antibiotic and antifungal activities, AgNPs have been integrated into bandages, biomedical devices, food storage apparatuses, textiles, cosmetics, and material surface coatings [4,5]. Additionally, their plasmonic capabilities have made AgNPs attractive for biomedical applications including bioimaging, optical sensor development, drug delivery vehicles, and as a mechanism to harvest photonic energy [5,6].

As NP prevalence increases, a corresponding rise in human contact also occurs, whether intentionally via therapeutics or via unintentional introduction. Furthermore, AgNPs have been linked to potential health concerns, with exposure resulting in significant cytotoxicity [7]. In addition to cellular death, AgNP exposure has been shown to induce numerous bioeffects including augmented oxidative stress, activation of the inflammatory and immune responses, alterations to signal transduction pathways, and genetic modifications [8,9,10,11]. For example, a study by Comfort et al. demonstrated that, even at dosages in the pg/mL range, AgNPs were able to disrupt normal cell homeostasis by augmenting cellular stress, activating Akt and Erk signal transduction pathways, stimulating pro-inflammatory cytokine secretion, and regulating gene expression [9]. In a recent study by Khan et al., 55 nm AgNPs were shown to penetrate the blood brain barrier and induce oxidative stress pathways and production of several proteins involved in neurodiseases [12]. In addition to cellular death, AgNP exposure has been linked to disruption of the human microbiota, autoimmune diseases, and environmental health concerns, highlighting the far-reaching implications of AgNP exposure [13,14] Given the potential for serious health and environmental complications, it is necessary that the cellular response to NP exposure be fully characterized and understood. However, cataloging the safety of NPs is a major challenge as bioresponses are dependent upon a unique combination of physicochemical properties, such as primary size, surface chemistry, core composition, and morphology [15,16]. This means that each AgNP set has the potential to induce a distinctive cytotoxic profile within a biological environment.

In addition to NP physicochemical properties, the biological exposure environment also plays a significant role in how cells recognize and interact with NPs. For example, the type of cell, available surface receptors, and surrounding environmental factors all contribute differently to the NP protein corona and nano-cellular interface, and the subsequent cytotoxic outcomes [17,18]. In a recent study, it was determined that two different hepatic cell models uniquely interacted and displayed differential cytotoxicity following exposure to identical gold nanoparticles [17]. Currently, NP safety assessments are carried out in either cell-based in vitro or animal-based in vivo models. While in vitro systems possess experimental flexibility, are cost-effective, and have high throughput capabilities, they significantly lack physiological relevance [19]. The discrepancy between these models has resulted in poor correlation of nanotoxicological results, with NPs inducing more diverse and greater cellular responses in vitro versus in vivo counterparts [20,21,22,23,24].

One approach to overcome the poor correlation between cell and animal systems is to develop enhanced biological models, which retain the in vitro advantages but incorporate key in vivo influences to produce a more realistic and relevant NP exposure scenario. One example is the inclusion of dynamic fluid movement, which can mimic the biological transport mechanisms of cardiovascular flow, and thereby include a more accurate prediction of NP transport. Initial dynamic studies identified that lateral fluid flow altered the balance between NP diffusion and sedimentation, thereby modifying dosimetry and resultant bioresponses [25,26]. A study by Fede et al. demonstrated that under static conditions gold nanoparticles had a higher rate of sedimentation and cytotoxicity than when exposure occurred within a dynamic flow environment [26]. The incorporation of fluid dynamics is critical in biological models that experience high rates of fluid motion, such as the blood brain barrier and vascular systems [27,28]. Additionally, exposure within a dynamic environment has been found to alter the protein corona, the recognized biological identity of NPs [29,30]. While several studies have examined the role of fluid dynamics on adherent cell models, to date, no work exists on how fluid dynamics influences the NP cellular association, stress, and toxicity within a suspension cell model.

The goal of this study was to elucidate and characterize the impact of dynamic flow on AgNP behavior, deposition, and biological response in both an adherent (A549) and a suspension (U937) human cell model. Lung cells, A549, were specifically chosen as inhalation is a primary form of NP exposure and A549s have been widely utilized in nanotoxicological evaluations [31]. The monocytic U937 cell line was of particular interest as monocytes are primary responders to foreign material in the body and trigger systemic oxidative and inflammatory responses [32]. Fluid flow within the cellular systems was generated using a peristaltic pump, which produced a tube-side linear velocity of 0.2 cm/s, equivalent to known capillary rates [33]. This study demonstrated that the presence of dynamic flow impacted AgNP deposition and the ensuing cellular responses for the experimental 50 nm AgNPs. In particular, several intracellular oxidative stress pathways were activated within both the A549 and U937 models. Interestingly, the adherent and suspension cells responded inversely to dynamic flow, with the A549 cells correlating to lower AgNP interactions and stress with fluid dynamics. It is hypothesized that, under the influence of fluid dynamics, the suspension U937 cells experienced augmented interactions with flowing AgNPs, thereby inducing more significant oxidative stress outcomes.

## 2. Materials and Methods

### 2.1. AgNP Characterization

The experimental 50 nm polyvinylpyrrolidone (PVP)-coated AgNPs were purchased from nanoComposix in concentrated solution form. nanoComposix high quality control guarantees that their products are endotoxin free. To minimize AgNP modifications over the duration of experimentation, the particles were stored at 4 °C in the dark. Transmission electron microscopy (TEM) analysis was performed on a Hitachi H-7600 to verify primary particle size and morphology.

For the remaining AgNP characterization assessments, the particles were diluted to a concentration of 25 µg/mL in the denoted fluid. The unique spectral signatures of these particles were visualized via ultraviolet visible (UV-Vis) spectroscopy on a Synergy 4 BioTek microplate reader. Degree of agglomeration was measured through dynamic light scattering (DLS) on an Anton Paar Litesizer 500. The Anton Paar Litesizer was also used to run zeta potential analysis in order to determine particle surface charge. 

### 2.2. Mammalian Cell Culture

Both human cell lines utilized in this study, the lung epithelial A549 and the monocytic U937, were purchased from American Type Culture Collection. The cells were maintained in RPMI 1640 media supplemented with 10% fetal bovine serum (FBS) and 1% antibiotics. The A549 cells, which are adherent in nature, were cultured on tissue-culture-treated petri dishes. The U937s were grown in suspension in T-75 tissue culture flasks. Both cells lines were sub-cultured every 3 to 4 days, as necessary.

### 2.3. Establishment of Dynamic Environment

A multi-channel peristaltic pump (Ismatec – Wertheim Germany, model #ISM939D) was used to establish dynamic flow, as previously demonstrated [25]. Both inlet and outlet ends of the tubing, with a 1/16 inch inner diameter, were secured into individual wells of a 24-well plate to generate unilateral flow throughout the cellular systems. Prior to experimentation, protein-rich media were run through the system to coat the tubes, thereby minimizing unintentional AgNP binding. For A549 experimentation, the pump was primed with either untreated or AgNP-dosed media to ensure that media levels were equivalent for both static and dynamic conditions. As U937 cells are suspension cells by nature, the pump was also primed prior to experimentation, but additional cells were included in order to maintain a constant cellular density with respect to media volume. To ensure no alterations in temperature, the pump and tubing were stored within the incubator. During experimentation, the pump operated at a rate that produced a tube-side linear velocity of 0.2 cm/s, which was specifically chosen to match physiological rates within capillaries [33].

### 2.4. Nano-Cellular Association

To determine AgNP interaction with the biological systems, cells were seeded into 24-well plates (2 × 10^5^ A549 cells/well or 1.5 × 10^5^ cells/mL for U937) and incubated overnight. The denoted cells were exposed to 15 µg/mL of the 50 nm AgNPs, under static or dynamic conditions. This AgNP concentration was selected as it was large enough to provide a significant detection signal but did not induce a strong cytotoxic response. Following 24 h exposure, the AgNP-containing media was collected; the U937 cells were removed via low-speed centrifugation, which removed the cells but not the suspended AgNPs. The supernatant then underwent UV-VIS analysis to determine the final NP concentration, using previously generated concentration curves [34]. A separate calibration curve was generated for static and dynamic conditions to account for any non-specific binding that may have occurred under dynamic conditions. The difference between this final concentration and the administered 15 µg/mL dosage was the AgNP amount associated with the cells.

### 2.5. Cellular Viability

Cell viability was determined using the CytoTox 96 Non-Radioactive Cytotoxicity Assay (Promega), which measures the production of lactate dehydrogenase (LDH). A549 cells were plated into 24-well plates at a concentration of 2 × 10^5^ cells per well. U937s were seeded into 24-well plates at a density of 1.5 × 10^5^ cells/mL, as they were in suspension. The following day, the cells underwent exposure to the 50 nm AgNPs at 0, 5, or 25 µg/mL, under either static or dynamic conditions. After 24 h, the media were removed and centrifuged to remove AgNPs and suspension cells. The supernatant then underwent LDH evaluation in accordance with the manufacturer’s instructions. A positive toxicity control, in which all the cells were lysed, was used to normalize data and determine degree of cytotoxicity.

### 2.6. Intracellular ROS Production

Intracellular stress was assessed by measuring reactive oxygen species (ROS) levels following AgNP exposure under the denoted conditions. A549 and U937 cells were seeded in 24-well plates at 2 × 10^5^ cells per well or 1.5 × 10^5^ cells/mL, respectively, and returned to the incubator until the following day. Next, the cells were washed and incubated with the DCFH-DA probe (Thermo Fisher Scientific) for 30 min, washed again, then dosed with the AgNPs at 0, 5, or 25 µg/mL within either a static or dynamic environment. After 24 h incubation, the ROS levels were measured via fluorescent analysis using a Synergy 4 BioTek microplate reader. Untreated cells under static conditions served as the negative control for normalization and hydrogen peroxide-dosed cells served as a positive control.

### 2.7. Glutathione Levels

The ratio of reduced glutathione (GSH) to oxidized glutathione (GSSG) is a metric to quantify intracellular oxidative stress [35]. The experimental cells were seeded and exposed to 0, 5, or 25 µg/mL AgNPs as described above for ROS. After 24 h, the cells were then washed, collected, and lysed in a non-denaturing lysis buffer. Intracellular GSH and GSSG levels were quantified using the GSH/GSSG Ratio Detection Assay Kit from Abcam, using a Synergy 4 BioTek microplate reader. GSH/GSSG ratios were calculated and normalized against an untreated, static control for each cellular system. Hydrogen peroxide exposure served as a positive control for known stress activation.

### 2.8. Activation of Intracellular Targets

A549 and U937 cells were seeded in a 24-well plate at 2 × 10^5^ cells per well or 1.5 × 10^5^ cells/mL, respectively. The following day the cultures were dosed with 0, 5, or 25 µg/mL AgNPs, under either static or dynamic conditions. After 24 h, the cells were washed, collected, and lysed in a non-denaturing lysis buffer. Activation of intracellular p53 and NFκB were determined by quantifying phosphorylated levels of these targets. In both cases, phosphorylated levels were normalized by the total p53 and NFκB amounts, quantified in parallel. Phospho and total levels of p53 and NFκB were assessed using protein-specific PathScan ELISA kits from Cell Signaling Technology, in accordance with the manufacturer’s instructions. Experimental activation levels were normalized against untreated, static controls. Hydrogen peroxide-exposed cells served as a positive control as they are known to induce numerous intracellular stress pathways.

### 2.9. Inflammatory Cytokine Secretion

The inflammatory response to AgNP exposure was determined by quantifying the production of the cytokines interleukin-1β (IL-1β) and tumor necrosis factor-α (TNF-α). While a panel of cytokines are secreted within an inflammatory response, IL-1β and TNF-α were selected as representative protein markers for this assessment. A549 and U937 cells were seeded in 24-well plates at 2 × 10^5^ cells per well or 1.5 × 10^5^ cells/mL, respectively, incubated for 24 h, then exposed to AgNPs under the stated conditions. Hydrogen peroxide exposure served as a positive control as it is known to induce numerous intracellular stress pathways. After 24 h exposure, the media were collected, AgNPs/U937s were removed via centrifugation, and the supernatant underwent analysis for extracellular IL-1β and TNF-α levels using protein-specific ELISAs (Thermo Fisher Scientific, Waltham, MA, USA), in accordance with the manufacturer’s directions.

### 2.10. Data Analysis

All data are presented as the mean ± the standard error of the mean. All experimentation was carried out with three independent trials. For cellular association analysis, a one-way ANOVA with Bonferroni post-test was run using GraphPad Prism to determine statistical significance between static and dynamic conditions. For the remaining experimentation, a two-way ANOVA with Bonferroni post-test was run using GraphPad Prism, with * and † indicating significance compared to the untreated control and between static and dynamic conditions, respectively.

## 3. Results

### 3.1. AgNP Characterization

Prior to cellular exposure, the AgNPs underwent numerous characterization assessments to quantify the unique physicochemical properties associated with this NP stock. As small deviations in NP properties have been correlated to differential bioresponses, it was necessary to first characterize the experimental NPs [36]. A representative TEM image of the 50 nm AgNPs is shown in Figure 1 and demonstrates a uniform, spherical morphology. Using multiple images, the primary particle size was determined to be 52.6 ± 6.9 nm. The uniformity of the AgNP stock was further verified through spectral analysis, which was comprised of a single, sharp peak (Figure 1B). When dispersed in media, there is a very slight right-shift in the spectral image, which can be attributed to the formation of a protein corona or minimal particle agglomeration.

In addition to verifying the primary particle size, the agglomerate sizes were determined in both water and media (Table 1). As all NPs will agglomerate when in solution, it is important to assess the extent of agglomeration as it can impact mechanisms of biotransport [37]. The AgNPs displayed minimal agglomeration in water with an increased effective diameter in media, due to the formation of a protein corona [38]. As the NPs had a PVP surface coating, which is known to promote particle stability in solution, the small extent of aggregation was expected [39]. Additionally, the surface charge was assessed using zeta potential measurements. The stock AgNPs displayed a negative surface charge, approximately -30 mV. Following the formation of a protein corona in media the charge shifted to -10 mV, as the outermost NP surface was covered in proteins which innately have a slight negative charge [38].

A peristaltic pump was used to generate fluid dynamics within the cellular system. Pump utilization afforded the opportunity to assess the impact of fluid dynamics on both AgNP characteristics and the nano-cellular interface. The targeted pump flow rate produced a linear velocity of 0.2 cm/s within the tubing, aligning with known capillary values. This means that the velocity across the cells was orders of magnitude less, aligning with the diffusion-based transport observed within tissue. Next, the AgNPs, suspended in either water or media, were added into an acellular dynamic system, circulated for several hours, and characterized in order to determine if the shear stress impacted key physicochemical properties. Previous reports have determined that dynamic flow is capable of modifying the protein corona [29,30], which could disrupt both AgNP agglomerate size and surface charge. As seen in Table 1, the low level dynamic movement had a negligible impact on both the extent of agglomeration and zeta potential measurements.

### 3.2. Dynamic Flow-Modified AgNP Cellular Interactions

Next, the influence of dynamic flow on AgNP deposition was investigated, for both the adherent A549 and the suspension U937 cell models. As seen in Figure 2, dynamic flow significantly altered NP association efficiencies, though in opposite modalities for these culture types. For A549 cells, the presence of lateral fluid flow decreased AgNP deposition by approximately 33%. The 50 nm AgNPs would have been associated with a significant rate of sedimentation, thereby producing a high number of AgNP–A549 interactions [35]. Under dynamic flow, the lateral convective movement kept the AgNPs in suspension, resulting in the observed deposition decrease. For the suspension U937 cells, the fluid dynamics, which counterbalanced downward sedimentation effects, resulted in augmented AgNP cellular association. As shown in Figure 2, when U937 underwent AgNP exposure under dynamic conditions, there was an approximate 66% increase in NP cellular binding.

### 3.3. AgNP-Induced Cytotoxicity Varied with Flow Condition

As AgNPs are known to induce mammalian cytotoxicity [8,9,10,11], the first endpoint assessed was cellular viability, as a function of flow condition (Figure 3). Starting with the A549s (Figure 3A), the lower AgNP dosage produced a slight cytotoxic response. However, following 25 µg/mL exposure, there was an approximate 40% loss of cell viability under static conditions. Within a dynamic A549 model, the cells still displayed AgNP dose-dependent cytotoxicity, though to a lesser degree, indicating that cellular damage was mitigated within a dynamic environment. 

The U937 cells also demonstrated a dose-dependent cytotoxicity response following AgNP exposure, as expected. However, when examining the U937 dynamic flow results, opposite trends were identified (Figure 3B). Following 25 µg/mL exposure, the U937s experienced a mild 12% toxicity under static conditions. With the inclusion of dynamic flow, this toxicity rate increased to approximately 25%, presumably due to amplified AgNP–U937 interactions. These viability results are in excellent agreement with the AgNP cellular association profiles, which for A549 and U937 cells identified a respective decrease and increase in nano-cellular associations. Moreover, any cellular changes to cytotoxicity were not directly related to the presence of fluid dynamics as shown in controls.

### 3.4. Intracellular Oxidative Stress Levels

As the presence of dynamic flow was able to modulate AgNP-dependent cytotoxicity, the next goal was to explore if this phenomena translated to intracellular responses, which precede cell death. Based on the dose-dependent toxicity analysis, an exposure of 5 µg/mL AgNP was selected, as it did not elicit a strong degree of cellular death. Detection of ROS was used to monitor intracellular oxidative stress levels, as it is a known cellular response following NP exposure and an early indicator of apoptosis [40,41]. In the absence of AgNPs, ROS production was equivalent for static and dynamic conditions for both models (Figure 4). These controls demonstrated that dynamic flow did not impact basal A549 or U937 stress levels.

As shown in Figure 4A, 50 nm AgNP exposure within a static A549 model demonstrated elevated ROS levels. In agreement with the cell viability data, under dynamic conditions the A549 stress response was significantly decreased. Following static AgNP exposure, U937 cells demonstrated a mild ROS response, indicating active intracellular oxidative stress. However, when U937 cells underwent the same AgNP dosage exposure under dynamic conditions, the resultant ROS levels were augmented by approximately 50%.

Intracellular glutathione exists in two states, reduced (GSH) and oxidized (GSSG). In normal, healthy cells, over 90% of all glutathione is in the GSH state. When cells are experiencing stress, the GSSG level rises; therefore, a decreased GSH/GSSG ratio is indicative of augmented oxidative stress [42,43]. As seen in Figure 4B, both A549 and U937 were experiencing stress following AgNP exposure under static conditions, due to GSH/GSSG ratios that were lower than the untreated control. Upon the introduction of fluid dynamics, GSH/GSSG ratios increased and decreased in the A549 and U937 cells systems, respectively, in agreement with the ROS data.

### 3.5. Activation of Oxidative Stress Protein Markers

Following identification of differential activation of ROS and glutathione responses within a dynamic environment, the next assessments targeted key proteins involved in the intracellular stress pathways. The goal was to elucidate whether p53 and NFκB were activated following AgNP exposure and whether the presence of dynamic flow modified the degree of this stress response. The targets of p53 and NFκB were specifically selected as these proteins have been previously shown to be induced following AgNP exposure in vitro [44,45]. As seen in Figure 5A, p53 was activated in both A549 and U937 systems following AgNP exposure. The phosphorylation of p53 mirror the ROS and glutathione results, with dynamic flow decreasing and increasing stress levels within the adherent and suspension models, respectively. When looking at NFκB activation, A549 showed both the AgNP-dependent stimulation and flow-dependent effects. However, U937 cells did not show any change in NFκB expression or activation following AgNP introduction, indicating that this pathway was not influenced by NP presence. Taken together, these results confirmed that exposure environment does influence intracellular oxidative stress proteins and pathways, though through cell-specific mechanisms.

### 3.6. Secretion of Pro-Inflammatory Cytokines Following AgNP Exposure In Vitro

AgNP-dependent acute stress induction has been linked to the activation of inflammatory responses in mammalian cells [46]. Once activated, an early inflammatory response is the production and secretion of pro-inflammatory cytokines, including IL-1β and TNF-α [47]. For untreated controls, the IL-1β and TNF-α levels were equivalent for static and dynamic conditions, indicating that fluid flow did not alter cytokine production (Figure 6). As A549 cells are alveolar epithelial, they only secrete a basal, low-level amount of cytokines. As shown in Figure 6, AgNPs did not elicit a significant inflammatory response under any condition. However, U937 monocytes, which are a part of the immune system, produced a robust array of cytokines following activation, including IL-1β and TNF-α. Regardless of exposure environment, 5 µg/mL AgNPs initiated a pro-inflammatory response, through upregulated cytokine production. Under dynamic conditions, a small increase in production levels was identified, indicating some degree of behavioral variance resulting from fluid flow conditions.

## 4. Discussion

The goal of this work was to elucidate the roles of cellular type (adhesion vs. suspension) and fluid dynamics in the AgNP cellular interactions and subsequent bioresponses, with a focus on oxidative stress markers. The results presented in this study demonstrated that both cellular and exposure characteristics significantly impacted the nano-cellular interface and the biological response. Therefore, each of these influences is discussed in greater detail below. Moreover, how this work supports the need and highlights implications for more physiologically relevant NP exposure models are included as points of discussion.

### 4.1. Influence of Fluid Dynamics on Different Cell Classifications

The addition of forced convective flow increased the biological relevance of the in vitro system, as all animal models are dynamic due to the presence of cardiovascular systems. By recreating a capillary-based velocity within the tubing, this model established diffusion-level transport across the A549 surface or alongside the U937s, representing tissue-level shear rates experienced in vivo [48]. This study determined that the addition of dynamic flow altered the NP transport profiles, as the lateral force counteracts the downward sedimentation and random diffusion which dominate under static conditions [49]. Firstly, when examining the impact of fluid flow, it was found that the utilized volumetric flowrate did not alter AgNP physicochemical properties (Table 1). These results suggest that all the elucidated cellular modifications were due to an altered nano-cellular interface and not the AgNPs themselves.

For both the adherent A549 and the suspension U937 cells models, the addition of dynamic flow was found to impact nano-cellular interactions, AgNP-dependent cytotoxicity, and subsequent oxidative stress responses. However, inclusion of dynamic flow in the exposure environment had opposite impacts on the A549 and U937 models, due to their differential growth behaviors. For the adherent A549 cells, dynamic flow reduced AgNP deposition, whereas nano-cellular associations were increased for the U937 model. In agreement with previous studies from the literature, the level of AgNP interaction with the surrounding biological environment dictated the degree of cytotoxicity and oxidative stress [50,51]. As such, when AgNP exposure occurred within a dynamic A549 model, the cytotoxicity and array of oxidative stress markers were assuaged. On the contrary, the higher rate of AgNP and U937 collisions when both were flowing in the dynamic model resulted in augmented toxicity and stress pathways.

Looking beyond toxicity, this study focused on numerous oxidative stress markers, including ROS, glutathione levels, p53, NFκB, and cytokine secretion. Taken together, these pathways highlight a multitude of differential intracellular stress responses. These endpoints were found to vary as a function of both exposure environment and cell model, mirroring the cytotoxicity results. This study is the first to our knowledge that has explored the role of fluid dynamics on the nano-cellular interface and oxidative stress within a suspension-based in vitro model, highlighting the variance in cellular behavior. Of particular importance is the ability of AgNPs and the exposure environment to influence the secretion of pro-inflammatory cytokines from U937 cells, which introduce the potential for a systemic inflammatory response and long-term health consequences [52].

### 4.2. The Need for Improved In Vitro Models

The results presented in this study highlight the necessity of developing in vitro models that are more biologically relevant and are able to better assess how NPs will behave within in vivo environments. The need for enhanced systems is further supported by the conflicting results and poor correlation between cell- and animal-based models with regards to nanotoxicity and the induction of intracellular stress and signaling pathways [20,21,22,23,24]. Discrepancies between these models are due in part to the fact that NP physicochemical properties and dosimetry are dependent upon environmental factors. For example, when NPs are dispersed in physiological fluids, such as interstitial and lysosomal, the rate of ionic dissolution and extent of agglomeration are significantly altered versus cell culture media [53,54]. Additionally, NP exposure within a three-dimensional in vitro model diminished the extent of cytotoxic responses versus two-dimensional models, owing to the need for NPs to translocate through the cell systems prior to internalization [55]. In agreement with this work, fluid flow has previously been found to modify cellular morphology, dosimetry, and resultant bioresponses [25,26,27]. 

In addition to further validating the need for more physiologically representative exposure scenarios, this work determined that the impact of fluid dynamics on NP dosimetry and dependent oxidative stress varied with cell system behavior. The results that emerged from this dynamic model better align with published in vivo data, in which AgNP exposure at lower dosages elicits only a minor stress response, without a large induction of cytotoxicity [56,57]. Therefore, this work indicates that the addition of dynamic flow increases the relevance of in vitro models and has the potential to improve predictive modeling capabilities. 

## 5. Conclusions

This study explored the role of cell adhesion properties and dynamic fluid movement on NP cellular interactions, cytotoxicity, and intracellular oxidative stress pathways following exposure to AgNPs. This work revealed that fluid flow modified the nano–bio interface but through differential mechanisms for adherent and suspension cells. For adherent cells, dynamic flow reduced AgNP contact and subsequent bioresponses due to a disruption of sedimentation. In contrast, cells that grow in suspension were associated with higher AgNP contact and oxidative stress activation within a dynamic exposure environment. The model implemented in this work, which incorporated dynamic flow, demonstrates how enhanced in vitro systems capture physiological influences and can thereby produce differential biological responses. Alterations to in vitro models also allow experimental flexibility and the ability to tailor design characteristics. For example, this study focused on low, diffusion-based flow that is associated with diffusion-level circulation. However, by switching to an endothelial cell model and increasing fluid flow rates, an arterial exposure system could be recreated.

## Figures and Tables

**Figure 1 antioxidants-10-00832-f001:**
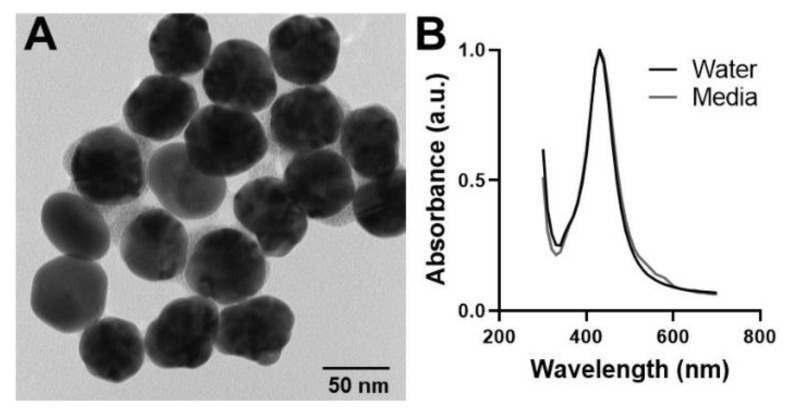
Characterization of the experimental 50 nm AgNP stock. (**A**) A representative TEM image of the 50 nm PVP-coated AgNPs is shown. This image verifies spherical morphology and an even particle size distribution. (**B**) The spectral profiles for the 50 nm AgNPs in both water and media demonstrate a sharp, single peak, further verifying particle uniformity.

**Figure 2 antioxidants-10-00832-f002:**
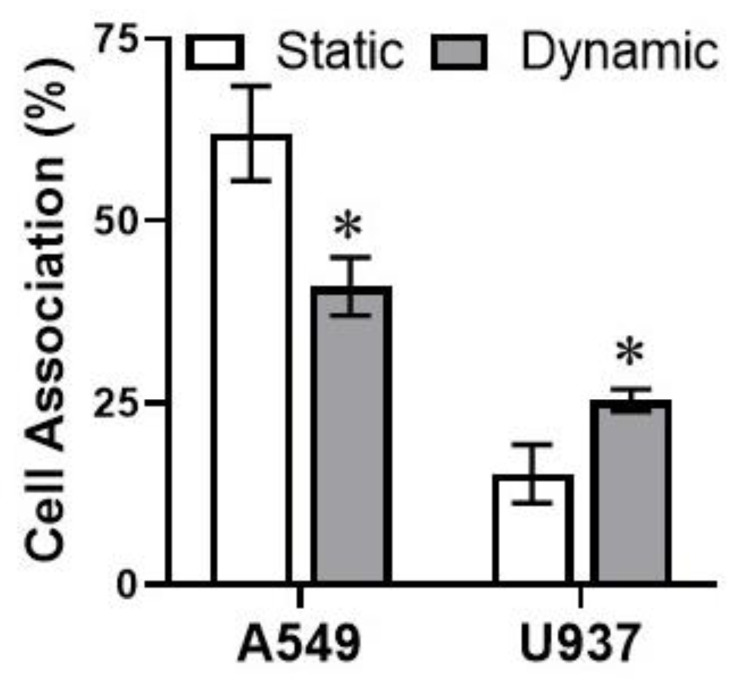
AgNP cellular association varies as a function of flow environment and cell model. Following 24 h exposure, AgNP interactions with the adherent A549 or suspension U937 models were measured within both static and dynamic environments. * indicates significance between static and dynamic conditions. *n* = 3, *p* < 0.05.

**Figure 3 antioxidants-10-00832-f003:**
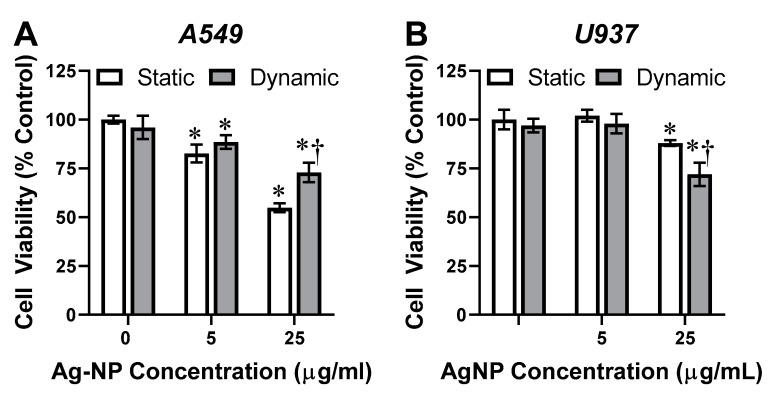
Cellular viability following AgNP exposure. Following 24 h AgNP exposure within either (**A**) A549 or (**B**) U937 models the cellular viability was assessed. These results indicated that within both systems viability was a function of AgNP dosage and flow condition. * and † indicate statistical significance compared to the untreated control and between static and dynamic conditions, respectively. *n* = 3, *p* < 0.05.

**Figure 4 antioxidants-10-00832-f004:**
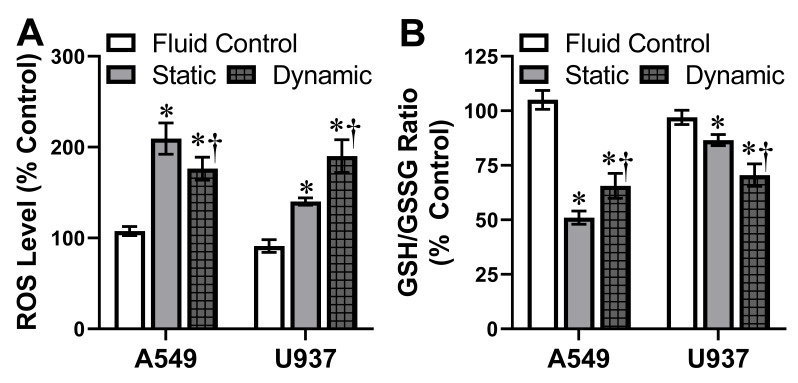
AgNP-induced intracellular oxidative stress levels. Both (**A**) ROS and (**B**) GSH/GSSG levels were used to monitor A549 and U937 intracellular oxidative stress following 5 µg/mL AgNP exposure for 24 h. Experimentation was carried out under both static and dynamic conditions. Dynamic flow without AgNP conditions were also included as fluid control. Oxidative stress levels were found to be a function of cell type and flow condition. * and † indicate statistical significance compared to the untreated control and between static and dynamic conditions, respectively. *n* = 3, *p* < 0.05.

**Figure 5 antioxidants-10-00832-f005:**
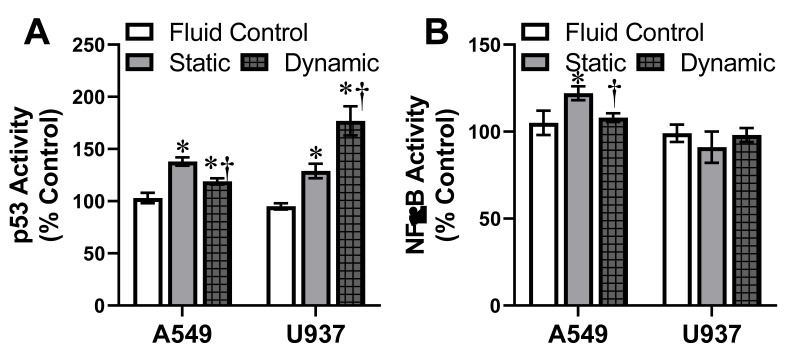
Activation of oxidative stress protein markers. Following exposure to 5 µg/mL AgNP, under both static and dynamic conditions, the A549 and U937 cells underwent evaluation for (**A**) p53 and (**B**) NFκB activation. Activation was determined by quantifying the phosphorylated state of target intracellular proteins, normalized by the total amounts of that protein. Activation of chosen oxidative stress pathways varied with cell type and exposure environment. Dynamic flow without AgNP conditions was also included as fluid control. * and † indicate statistical significance compared to the untreated control and between static and dynamic conditions, respectively. *n* = 3, *p* < 0.05.

**Figure 6 antioxidants-10-00832-f006:**
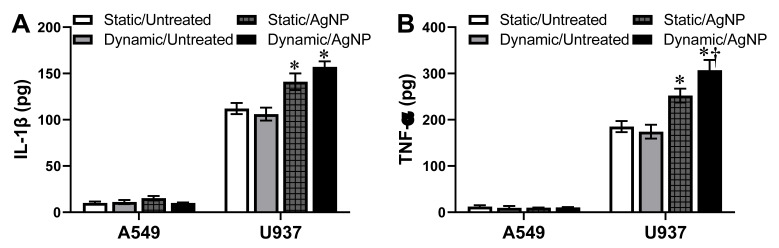
Secretion of pro-inflammatory cytokines following AgNP stimulation. The alveolar epithelial A549 and monocytic U937 cell lines were exposed to the denoted combinations of static/dynamic conditions and 5 µg/mL AgNPs. After 24 h the media were recovered and evaluated for levels of the pro-inflammatory cytokines of (**A**) IL-1β and (**B**) TNF-α. * and † indicate statistical significance compared to the untreated control and between static and dynamic conditions, respectively. *n* = 3, *p* < 0.05.

**Table 1 antioxidants-10-00832-t001:** Characterization of the AgNPs under static and dynamic conditions.

Flow Condition	Agglomerate Size (NM)		Zeta Potential (MV)	
	Water	Media	Water	Media
Static	78.6 ± 3.0	89.3 ± 2.2	−30.4 ± 1.7	−9.7 ± 0.6
Dynamic	76.7 ± 2.4	90.8 ± 2.9	−32.0 ± 2.4	−10.5 ± 1.0

## Data Availability

All data are contained in this article and Appendix A.

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
