# Peer review of "Characterizing the Role of Biologically Relevant Fluid Dynamics on Silver Nanoparticle Dependent Oxidative Stress in Adherent and Suspension In Vitro Models"

_antioxidants, 2021, doi:10.3390/antiox10060832_

Round 1

Reviewer 1 Report

In their manuscript, Burns et al. tested the effects of media flow on silver NP toxicity and cell stress responses. They exposed an adherent and a suspension cell model to static conditions or dynamic media flow. They show that dynamic flow affects NP-cell association, NP toxicity and cell stress responses, including oxidative stress, stress related transcription factor activity and cytokine secretion. The study is well executed and well written, the results are sensible and easy to follow.

However, there are some aspect of the study that need clarification.

A549 might not be the best adherent cell model for this type of study. In vivo, these cells form a pulmonary epithelium and are separated from the blood flow by an endothelial cell layer that surrounds the capillaries. Moreover, following inhalation exposure, the pulmonary epithelial cells would be exposed to airborne NPs from the apical side and indirectly to the blood blood flow on the basal side. As the authors’ claim they wanted to establish a more realistic cell model, a (differentiated) endothelial cell model would be a better choice to study the effect of physiological flow rates within the capillaries. Moreover, the reference the authors cite for »significant fluid dynamics [28]” in the alveolar region described airflow, not liquid. Based on this, authors’ claims, explanations and references should be revised and replaced.

Authors determined the amount of NP-cell association by measuring the quantity of NPs that remained in the cell culture media following exposure, so the quantity of NPs that did not get adhered. The experimental setup for study of dynamic flow involves a lot of plastic tubes, which were probably not treated to be “low binding”, thus the dynamic flow samples were exposed to significantly greater potentially adherent surfaces compared to static samples. The authors should confirm the NPs do not adhere also to the plastic. Also, since the authors have access to TEM, association (and potentially) internalization of NPs should be confirmed microscopically or determine the quantity of NPs associated with cells directly (instead of measuring the medium).

In Figure 2, the authors’ show that in U937 cell model, there was less NP-cell association in static conditions compared to dynamic. In static conditions, both cells and NPs would sediment to the bottom of the well in 24 h incubation period, bringing cells and NPs in close contact. How can association be lower in such conditions compared to dynamic conditions, where cells and NPs remain in suspension? How were the cells washed in order to remove unbound NPs from cells?

While the dynamic flow adds to the relevance of certain cell models, it is not applicable in general (eg. a great majority of cells are inside different tissues that are not exposed to such medium flow but have a much slower, passive exchange of substances). The description of the applicability and relevance of the established model should thus be more concrete (e.g. for studying the interactions of cells and NPs in blood …)

Reviewer 2 Report

Authors study the effects of dynamic flow on AgNPs behaviour, deposition, and biological response in an adherent alveolar (A549) and a suspension monocyte (U937) human cell models. They demonstrated that the dynamic flow reduced AgNPs deposition and oxidative stress markers in the adherent cells whereas the suspension U937 cells, due to increased frequency of contact, were associated with higher AgNPs interactions and intracellular stress under fluid flow exposure.

I recommend the minor revision of the manuscript.

  • Materials and Methods:

2.4. Nano-Cellular Association:

-     Why the author uses the concentration of AgNPs 15 µg/ml when they further used 5 µg/ml or 25 µg/ml?

  • Correct the unit of the cell concentration - 2x105 A549 cells per well.

2.5. Cellular viability

- Specify the concentration range of the AgNPs which were used for the viability evaluation.

- What the authors used as a positive toxicity control that was used to normalize data and determine degree of cytotoxicity?

2.6. Intracellular ROS production

- Specify the concentrations of the AgNPs which were used for the ROS evaluation

- According to my opinion, the authors performed the experiment incorrectly. First, the cells were seeded, after 24 hours the cells were incubated with probe and thereafter the AgNPs were applied on 24 hours and the oxidative stress were evaluated?

2.7. Glutathione levels / 2.8. Activation of intracellular targets

- Specify the concentration range of the AgNPs which were used for the evaluations.

- Specify the composition of the non-denaturing lysis buffer.

Results:

  • I recommend to add controls that are used for the significance evaluation into the graphs.

Supplementary data:

Which controls did the authors used for the evaluation of statistical significance? I thought that this are the controls which the authors used for evaluation of statistical significance of AgNPs treated cells.

Reviewer 3 Report

The authors present a study on the, in vitro, effects of AgNP on human cell lines. Whilst this has been extensively studied, elsewhere, the authors have added flow dynamics to provide an environment closer to the physiological barriers mentioned. The article is interesting however, there are a number of points that should be addressed before a final decision is made:

  1. There appears to be no consideration of potential biological contamination within the work. No effort has, apparently, been made to address possible bioburden, endotoxin, or other potential pyrogens within the materials. As the main output of this work is “inflammatory” responses, it must be considered prior to further consideration.
  2. None of the experiments appear to have any positive controls within them. Why have none been included to ensure that the cell lines are responding in an expected way? That is a very strange omission.
  3. There appears to be no consideration of inclusion of inhibition/enhancement controls. Although samples have been centrifuged to “remove nanoparticles” can that be assured? There should be data showing that the nanoparticles have been removed from the matrix. Appropriate inhibition/enhancement controls should have included incubation of nanoparticles with positive controls (which do not appear to have been included) to ensure there is no loss or enhancement of signal in the respective assays, particularly the cytokines secretion. Metallic NP are well known to cause these effects (such as binding to enzymes and inactivating them in in vitro assays) and, at present, it is unclear if the results consider this.
  4. What is the justification for a 24-hour exposure? Oxidative stress, and resultant cellular effects, are dynamic so why are there no earlier time points. Similarly, for cytokine analysis, the temporal responses of cells are much more important than what has been collected over a 24-hour period – this is vital to differentiate between direct NP effects and possible autocrine effects of inflammatory cytokines.
  5. Why were only IL-1b and TNFa measured? IL-6 is a very important cytokine that acts via both cis- and trans-signalling and can induce the secretion of other bioactive molecules. Additionally, AgNP are known to cause cellular effects, such as perturbation of mitochondrial membrane polarisation, that can affect intracellular ATP levels and induce inflammasome activation. Why was this not considered here? Given that IL-1b secretion is included, what about IL-18? What is the mechanism of induction of these cytokines?
  6. Protein Corona is mentioned but not assessed; given that the authors suggest that it may affect the characteristics of the AgNP, it should be considered.
  7. Why were only U937 used for the interaction analysis? How do the authors know if the particles were particles inside or on the membrane of the cells? As the main difference here is the flow angle, surely this is important as it shifts the exposure-response dynamics.
  8. One of the primary routes of toxicity with AgNP is the release of silver ions – where any present in the systems used here?
  9. It is unclear how the concentrations used in the exposures where arrived at; are they based on actual exposures that are seen in the population? This must be clarified and justified.
  10. Is the shift in the spectra due to corona or agglomeration? Considering the importance of both, this should be clarified.
  11. Explain the change in surface charge in media – how can a more negative charge in proteins adhered make something more positive (line217)? This sentence does not make sense in it’s current form; if more negatively charged proteins associate with the surface of the AgNPs, then the surface charge would become more negative and not less negative. Also, what is the rationale for association of negatively charged proteins with a negatively charged surface? This must be explained as it cannot be due to simple electrostatic interactions.

Reviewer 4 Report

Dear Authors,

After the review process, I have several comments: your should include more numerical data in the abstract; you should include more new data, in the introduction, about in vitro studies and AgNPs.For example, you could include data about in vitro human microbiota response to exposure to silver nanoparticles; you should include references in all Materials and Methods sections; you should insert a conclusion section, at the end of the paper.

Best regards.

Round 2

Reviewer 3 Report

I am pleased that the authors have considered the comments offered. Whilst I do not agree with a number of points in the rebuttal, the manuscript has improved significantly from the first draft and would now be suitable for acceptance.

Reviewer 4 Report

No supplementary comments.